# Cross-Cultural Validation of the Portuguese Version of the Quality of Oncology Nursing Care Scale

**DOI:** 10.3390/cancers16050859

**Published:** 2024-02-21

**Authors:** Pedro Gomes, Susana Ribeiro, Marcelle Silva, Paulo Cruchinho, Elisabete Nunes, Carla Nascimento, Pedro Lucas

**Affiliations:** 1Nursing Research, Innovation and Development Centre of Lisbon (CIDNUR), Escola Superior de Enfermagem de Lisboa, 1600-190 Lisboa, Portugal; pedro.gomes@campus.esel.pt (P.G.); marcellemsufrj@gmail.com (M.S.); pjcruchinho@esel.pt (P.C.); enunes@esel.pt (E.N.); carla.nascimento@esel.pt (C.N.); prlucas@esel.pt (P.L.); 2Escola de Enfermagem Anna Nery, Universidade Federal do Rio de Janeiro, Rio de Janeiro 20211-130, Brazil

**Keywords:** oncology nursing, quality of healthcare, health management, psychometrics, validation study

## Abstract

**Simple Summary:**

Patients are the center and focus of care delivery, but frequently, their perceptions are overlooked. Patients are vital to promoting optimal oncology nursing care and should be considered in order to provide patient-centered holistic care. The Quality of Oncology Nursing Care Scale was translated and validated cross-culturally for the Portuguese context. In Portugal, there is no instrument that evaluates the quality of nursing care provided to cancer patients, so our study fills this major gap in the scientific evidence. This scale should be used by nurses, nurse managers, and researchers that strive to improve patient, nurse, and organizational outcomes.

**Abstract:**

Background: Quality assessment in oncology nursing care has been a growing topic in the literature, gaining relevance as oncological nursing care becomes more complex as the science progresses. However, there are no instruments that assess the perception of the quality of oncology nursing care from the point of view of patients for the Portuguese population. Thus, the cross-cultural translation and validation of the Quality of Oncology Nursing Care Scale (QONCS) was performed for the Portuguese context. This instrument allows nurses to assess patients’ self-perception of the quality of nursing care provided in an oncological setting. It also allows researchers to compare the results obtained internationally with the application of this scale. Methods: This is a methodological study, with two distinct phases: the first corresponded to the translation and cultural adaptation of the scale to the Portuguese context, and the second consisted of the psychometric validation of the QONCS, which included factor analysis and the evaluation of the psychometric properties of the instrument. We obtained responses from 402 patients from a Portuguese oncology hospital. Results: The Portuguese version of the Quality of Oncology Nursing Care Scale (QONCS_PT) consists of 34 items inserted into a tetra-factorial model, which explains a total variance of the instrument of 69.8%. A Cronbach’s alpha of 0.93 was obtained for the complete instrument. Conclusions: QONCS_PT has a competent and reliable structure. The scale’s validity was assured and can be used in the Portuguese population, as it is useful for direct care provision but also for researchers and managers.

## 1. Introduction

In Europe, cancer is the second-most-common cause of death and morbidity, with more than 3.7 million new cases and 1.9 million deaths reported every year [1]. Cancer care has consistently become more specialized as the science advances, with oncology nurses striving to ensure the safe delivery of cancer treatments, optimize patient’s quality of life, coordinate nursing care delivery, improve outcomes, and mitigate cancer’s impact on patients, families, communities, and populations. Nurses who specialize in oncology care have a high impact by promoting healthier lifestyles, supporting cancer symptom management, and fostering early detection and diagnosis [2]. As oncology nursing care is becoming more complex, it is essential to assess its quality and how patients perceive quality care. Quality can be measured in different ways and should consider the specific characteristics of the organizational environment [3].

The United States Centers for Medicare & Medical Services (2024) defines quality measures as “tools that help us measure or quantify healthcare processes, outcomes, patient perceptions, and organizational structure and/or systems that are associated with the ability to provide high-quality health care and/or that relate to one or more quality goals for health care” [4]. For the Agency for Quality Healthcare Research and Quality, a quality measure is “a mechanism to assign a quantity to quality of care by comparison to a criterion” [5]. To evaluate quality measures, institutions collect information about their performance, asking patients to evaluate their services for a given quality measure [6]. In the field of cancer care, this information must be widely disseminated to support patients’ decision making [7].

Quality measures can be classified into three types: structure measures, process measures, and outcome measures. Structure measures assess the characteristics and infrastructures of a given healthcare organization that are relevant to its ability to provide healthcare, such as the ratio of providers to patients, and whether the healthcare organization uses medication order entry systems. Process measures can inform patients about the healthcare that they can expect to receive for a given health condition and can contribute to improving healthcare. The number of patients who receive a certain treatment for a certain disease is an example of a process measure. Outcome measures reflect the impact of healthcare services or interventions on patients’ health status, such as mortality, readmissions, effectiveness of care, waiting times, and patient experience [6].

The patient’s perspective is currently seen as a valuable source of knowledge on health and illness and perceives their experiences and actions in resolving any identified problems as representing a central objective in continuous quality improvement programs [8]. Although the healthcare sector has been negligent in measuring the effects of its activity on reported outcomes and experiences, collecting this knowledge in a systematic and useful way for decision making has an important role in the transformation towards people-centered systems in healthcare [9].

One of the ways to evaluate patients’ experiences and their needs is through Patient-reported Outcomes Measures (PROms) and Patient-reported Experience Measures (PREMs). PRO is an umbrella term that includes all subjective patient outcomes, such as pain, fatigue, depression, wellbeing (physical, functional, and psychological), satisfaction with treatment, health-related quality of life, and physical symptoms, such as nausea and vomiting, without the interpretation of health professionals [10]. PROMs allow the measurement of patients’ opinions about their symptoms, functional status, and health-related quality of life [11]. They allow a patient to determine the degree to which healthcare meets their personal and cultural expectations, respects the patient’s preferences, and takes personal values into account when care is provided [6].

PREMS measure patients’ personal experience of the healthcare they receive [11,12], excluding information about specific therapeutic interventions [13]. They are usually answered anonymously because people are typically reluctant to criticize those that they depend on for fear of reprisal [12]. They collect data about what happened from the patient’s perspective, rather than their general assessment [14]. Therefore, they differ from satisfaction questionnaires in that they focus on objective experiences, rather than objective opinions [15]. They can cover several dimensions, namely, accessibility, communication, continuity and coordination, shared decision making, and trust [9]. They are applicable to a variety of healthcare contexts to demonstrate trends in experience scores across healthcare services and systems over time [16].

Oncology nursing care should reflect the patient’s needs as a human being and be patient-centered, holistic, and up to date with the best practices, striving to continuously improve quality [17,18]. The introduction of PROMs and PREMs into routine care as part of quality improvement processes has multiple benefits: (1) the implementation of person-centered care; (2) increasing quality improvement and standardized care; and (3) providing an informative framework for research and policy development [11].

Healthcare organizations aim to deliver the best quality care for patients’ satisfaction [19,20]; however, despite the amount of instruments that assess the nursing care quality in specific specialty units, instruments that gauge patients’ perceived quality in oncology settings are still scarce. Only two have been found in the literature—the “Quality of Oncology Nursing Care Scale” [21] and the “Oncology Patients’ Perceptions of the Quality of Nursing Care Scale” [22]. However, only the first (QONCS) includes the assessment of spiritual care in oncology nursing care, therefore embracing holism in care.

In Portugal, the incidence and prevalence of cancer show no signs of slowing down anytime soon, and yet there is still no scale that aims to assess the perceived quality of oncology nursing care for the Portuguese population.

The scale has been translated and culturally adapted into six languages (Arabic, Czech, English, Greek, Korean, and Brazilian Portuguese) and is widely used to assess patients’ perceived quality of oncology nursing care by nurse managers and researchers [18,21,23,24,25].

The aim of this study was to transculturally translate and validate the QONCS for the Portuguese context. This scale evaluates the perception of the quality of oncology nursing care by patients. It comprises a total of 34 items and is composed of five distinct domains/subscales: “Being Supported and Confirmed”, “Spiritual Caring”, “Sense of Belonging”, “Being Valued”, and “Being Respected” [21].

## 2. Materials and Methods

### 2.1. Design

A methodological study was conducted with the objective of translating and culturally adapt the QONCS [21] for the Portuguese population.

The study included two phases: the qualitative phase, where the translation methodology of Beaton et al. [8] was used in order to obtain an adequate translation of the instrument; and the quantitative phase, where a factor analysis was conducted to validate the translated instrument’s accurate psychometric properties.

### 2.2. Sample and Data Collection

A non-probability sample of 402 patients voluntarily participated in this study. A data collection form was elaborated based on the one applied in the original research study [21] along with the translated scale to Portuguese. All forms were administered at a oncology hospital in Portugal. In order to maintain patients’ anonymity, the forms were deposited in a envelope, mixed with those from all the participant patients on the ward, and gathered bi-weekly by the authors of the study. All participants were admitted to medical surgical wards where oncological nursing care was delivered. Written formal approval was obtained from the head nurses and physicians on all the wards that were included in the study.

The inclusion criteria were based on three pillars: all patients should be aged 18 or older (as the scale was developed and tested for the adult population); all patients should be able to self-administer the form; and all patients should be admitted at the hospital and receiving oncology nursing care. Only one exclusion criteria was defined: inadequate proficiency in the Portuguese language.

### 2.3. Ethics Issues

Informed consent was obtained from all participants in the present study, and confidentiality and anonymity were guaranteed. Participation in the study was also voluntary and free of charge, respecting the principles of the Declaration of Helsinki [26], as well as the data protection of all those involved in the research project.

The authorization and opinion of the Ethics Committee were requested at the hospital where this study was applied. Along with the request to the Ethics Committee, a request to conduct the study was also sent to the Hospital’s Board of Directors. A favorable opinion was obtained from both parties (Opinion UIC/1439). 

All data were collected from May 2022 to September 2022, after approval from the hospital ethics committee and the board of directors.

### 2.4. Measurement Development

#### 2.4.1. Original Instrument

The Quality of Oncology Nursing Care Scale was originally developed and published by Andreas Charalambous and Theodoula Adamakidou in 2014 [21]. Its purpose arose from the need to proper assess patients’ perceived quality of oncology nursing care from an holistic perspective, where all dimensions of being in a holistic approach were taken into account. It is a 5-point Likert-type scale consisting of 5 dimensions and 34 items, with responses ranging from 1 (Strongly Disagree) to 5 (Strongly Agree). The five dimensions are “Being Supported and Confirmed”, “Spiritual Caring”, “Sense of Belonging”, “Being Valued”, and “Being Respected” [21].

The instrument is valid and reliable, with an excellent internal consistency (α = 0.95) and an explained variance of 68.53% for a penta-factorial solution [21].

The authors of the Quality of Oncology Nursing Care Scale gave us permission to translate and psychometrically validate the instrument for the Portuguese cultural context.

#### 2.4.2. Translation and Cultural Adaptation Process

In order to obtain an instrument with an accurate translation that fully grips and embraces measurement, semantic, and conceptual equivalence with the original scale, we followed the guidelines for translating instruments of Beaton et al. [27] and Sousa and Rojjanasrirat [28]. Semantic and conceptual equivalences were reached through the translation process as the measurement equivalence was grasped through the assessment of the psychometric properties [29].

Therefore, a back-to-back translation process was performed by two proficient bilingual translators, one of them with specific nursing knowledge. With the expert committee, a consensus pre-final version was achieved. A comprehensive pre-test is essential in this phase to detect patient’s intelligibility of the phrases [27]. Despite the importance of this step, the ideal sample size and which methodological approaches can be followed are still unclear in the literature. Regardless, most studies recommend using a sample size above 30–50 participants [27,30,31]. Some claim that 22 participants would suffice to detect 90% of the problems regarding intelligibility [30]. In this study, a non-probability sample of 52 patients was used in the pre-test. The intelligibility of the scale was investigated through cognitive debriefing after it was completed by patients. There is a need for guidelines to support the decision-making process in healthcare researchers with comprehensive information about the different methodological approaches that they can follow [32]. If more than 20% of the participants had doubts or questions concerning the scale, it would have to be re-analyzed, assessed, and even reformulated [28]. Around 96% of the participants (N = 50) did not encounter problems when filling out the form. However, two important points were raised by the remaining participants and one expert committee member. The remaining 4% of the patients felt that due to the SARS-CoV-2 pandemic restrictions, some items were difficult to gauge due the mandatory rule of “no visits” from family or significant others. Another important point was raised by a expert committee member concerning the use of the word “spiritual” in the instrument. Despite “spiritual” being a far broader term than “religious”, for the general older Portuguese population, it is easier to understand. Therefore, both terms were adopted jointly, using a “/”, for comprehension purposes.

### 2.5. Reliability and Validity

Validity is centered on the instrument’s ability to truly measure the studied construct and in its ability to gauge, or not, the construct [29,33,34]. Construct validity was reached through a exploratory factor analysis (EFA) and confirmatory factor analysis (CFA) [29,31,35]. Content validity was reached through the qualitative phase of the study translation process [27].

Reliability means that this questionnaire should consistently reflect the construct it is measuring [29,32].

### 2.6. Statistical Analysis

All data was analyzed using SPSS Statistics version 27.0 [36] and AMOS statistical software version 27.0 [37].

## 3. Results

### 3.1. Sample Data Analysis

The sample was composed of 402 patients admitted to a Portuguese hospital where oncology nursing care was delivered. Sexes were nearly evenly distributed. Data closely tie in with the Portuguese average sex distribution of 52.8% females and 47.2% males [19]. Most patients were 45 years old or older—the mean age was 57.1 and the median age was 57. Sturge’s formula was used to organize participants into different age brackets [38]. Head and neck cancer was the most predominant cancer, despite Portugal’s most prevalent type of cancer being colorectum cancer [39]. Other type of cancers included in the category “Other” were predominately hematological and gastroenterological (stomach and colon) cancer. More detailed data from the sample can be found in Table 1.

### 3.2. Reliability and Validity

A Cronbach’s alpha score of 0.93 was obtained for the Portuguese version of the scale, resulting in an excellent internal consistency, assuring the scale’s reliability for a 5-factor model questionnaire.

Content validity was obtained following the rigorous translation process mentioned in the qualitative phase of the study. Criterion validity was not considered, since there is no other oncology nursing management tool that assesses the patient’s perceived quality of oncology nursing care from an holistic perspective that the authors know of. Construct validity was determined following a factor analysis. Kaiser–Meyer–Olkin (KMO) and Barlett’s scores were calculated to assure the sampling adequacy [40,41] for the factor analysis—a KMO score of 0.928 and Bartlett’s score of <0.000 were obtained, assuring an excellent sampling adequacy and data correlation [42]. With an eigenvalue of 1 and scree plot criterion, a total variance of 73.15% was observed for a 5-factor matrix. Once the number of factors had been extracted, an orthogonal varimax rotation was performed in order to determine to what extent the variables saturate in these factors.

Since no items were included in the fifth factor, a forced reduction to four factors was performed. A KMO of 0.928 and a Bartlett sphericity test of <0.000 were again obtained, since no items were excluded from the scale. With an eigenvalue of 1 and the scree plot criterion, a total variance of 69.84% was observed for a 4-factor matrix. Once the number of factors had been extracted, an orthogonal varimax rotation was performed, in which all items had minimum recommended saturation values greater than 0.4 [31]. This time, all factors had associated items. Therefore, the final scale did not lose any of the 34 items from the original scale, However, they were reorganized, occupying different factors, namely, items 28 to 32, which became part of the first factor.

An assessment of the instrument’s reliability was re-carried out, where a Cronbach’s alpha score of 0.93 was obtained for the whole scale, revealing an excellent reliability. The internal consistency of each factor can be observed in Table 2.

### 3.3. Exploratory Factor Analysis

The first dimension of QONCS—Portuguese Version (QONCS_PT) essentially comprises the patients’ perception of the safety of the nursing care provided, the quality of communication established in the nurse–patient binomial, and the appreciation that the patient felt when nursing care was provided. This dimension comprises items 1 to 16 and 28 to 32 of the original QONCS scale, thus including 16 items from the “Being Supported and Validated” dimension, 4 from the “Being Valued” dimension, and 1 from the “Being Respected” dimension. Therefore, in order to be as faithful as possible to the conceptualization and nomenclature established by the original authors of the scale, the first dimension was named “Being Supported, Validated, and Valued”.

The second dimension will aim to incorporate spirituality in nursing care, adopting a holistic approach to care delivery. The second dimension attends to the patients’ perception regarding nurse’s interventions in their spiritual beliefs, that is, whether the nurse promoted and provided the necessary resources for the patient to practice their spirituality. Thus, the original name of the scale was adopted, adopting the nomenclature of “Spiritual/Religious Caring”, including items 17 to 22 of the original scale.

The third dimension comprises the incorporation of the family in the delivery of nursing care. The dimension adopts items 23 to 27 from the original scale, as well as its nomenclature, “Feeling Supported”. In this dimension, the aim was to listen to patients’ perceptions about nurses’ interventions in the involvement of family in the provision of care.

Lastly, the fourth dimension is intended to assess the degree of respect perceived in the nurse–patient relationship, that is, the level of respect that the patient felt from the nurse when providing care. The dimension includes items 33 and 34 of the original scale, as well as its nomenclature, “Being Respected”.

In Table 2, we can observe the eigenvalue of 1 score for each item in all four components of QONCS_PT.

### 3.4. Confirmatory Factor Analysis

A CFA was performed for the tetra-factorial solution with 34 variables using the maximum likelihood method. The original model was fitted to a sample of 402 patients, in which good fit values were obtained (X^2^ =7182.9; X^2^/gl = 2.273; RMR = 0.048; AGFI = 0.692; GFI = 0.730; CFI = 0.900; IFI = 0.900; RMSEA = 0.080; MECVI = 6.782; SRMR = 0.0519).

In order to enhance the adjustment of the model obtained in the preliminary CFA, 14 observations were eliminated (e26–e27; e31–e32; e17–e18; e10–e15; e4–e5; e15–e17; e22–e23; e13–e21; e20–e21; e10–e21; e2–e3; e3–e9; e5–e9; and e8–e18) whose D^2^ values were revealed to be outliers, culminating in their exclusion from the CFA.

Consequently, after eliminating these outliers, the global adjustment model was enhanced, and very good adjustment values were obtained (X^2^ =7182.9; X^2^/gl = 1.625; RMR = 0.047; AGFI = 0.780; GFI = 0.812; CFI = 0.952; IFI = 0.953; RMSEA = 0.056; MECVI = 5.159; SRMR = 0.0487) (Figure 1).

## 4. Discussion

The QONCS was translated into Portuguese, and the instrument proved to be reliable and valid for evaluating oncology nursing care.

Internationally, the scale has been officially translated and validated into Arabic [24] and Brazilian Portuguese [25]. However, the scale in its Arabic version has undergone changes regarding the way it presents the results for each dimension. That is, instead of the score being presented through an overall arithmetic mean or the values of each dimension, it is presented as the sum of the value for each item of the dimension. It is therefore impossible to compare the results. The Brazilian Portuguese version of the scale was not officially published.

The scale did not change its structure in any international cultural context [18,21,24,25,43] with five dimensions, other than in our research, in which a tetra-factorial model was adopted. The scale had outstanding results in all other cultural contexts and was proven to be reliable and valid. The highest internal consistency was present in the original version with 0.95 [21] and the lowest in the Arabic version with 0.88 [24].

The dimension “Being Supported, Validated, and Valued” had a high average score (4.76) for the Portuguese cultural context, translating into a perceived feeling of safety, appreciation, and support in patients when they were provided with nursing care. Comparatively, our study obtained an average score in line with the other European countries [18], while Republic of Korea [43] had the lowest average score (3.47).

A higher score reveals the greater quality of communication established between patients and nurses. For a nurse manager, this dimension is particularly important, since an adequate therapeutic relationship between a patient and nurse has a significant impact on the quality of nursing care provided. Establishing a therapeutic relationship leads to a reduction in the number of days spent in the hospital; increased patient and nurse satisfaction; increased patient autonomy [44]; and a reduction in suicide rates [45].

The dimension “Spiritual/Religious Caring” had a low average score for the Portuguese cultural context (2.21), revealing that customers did not feel supported in fulfilling their spiritual needs. However, this phenomenon is not only visible in the Portuguese cultural context, as it is evident in all countries where the scale was applied [18,21,24,25,43]. Spirituality is closely linked to the quality of care provided [46]. Spirituality is seen as a pillar in the provision of holistic care [47]; therefore, it is fundamental for a nurse manager and oncology nurse that this dimension is integrated into the oncology nursing care provided by the team they lead, to humanize care [48]. This concern is fundamental for nurses to prioritize in inpatient services. Providing spiritual support to patients, regardless of their religious denomination, is a way of reducing possible prejudices and errors in assessing spiritual needs.

As for the dimension “Feeling Supported”, it presented a relatively neutral average score (2.97) for the Portuguese cultural context. The result obtained is not particularly favorable when compared to the other countries in which the scale was applied, with Portugal being the country with the lowest average score and Cyprus [4] being the country with the highest average score (4.3). It is important to note that the questionnaires were applied during the pandemic caused by the SARS-CoV-2 infection, and considering the vulnerability of cancer patients, all visits to the wards were suspended, except for occasional exceptions caused by the deterioration of the clinical status of the patient; therefore, the results obtained in this dimension for the Portuguese cultural context may be improved, since the physical involvement of family members during the pandemic was mostly deferred in the provision of hospital nursing care.

Regarding the “Being Respected” dimension, favorable scores were obtained for the Portuguese cultural context, with an average score of 4.22. The score achieved shows that patients felt informed and respected when oncology nursing care was provided to them. Comparatively, the remaining countries had similarly high average scores, with Republic of Korea [9] having the lowest score (3.77) and Cyprus and the Czech Republic [18] the highest (4.5). In clinical settings, nurses can improve the perceived respect level by demonstrating an interest in patients’ points of view and recognizing and positively supporting their individual abilities. Particularly in older age patients, it was identified that a lack of active listening and encouragement was the main flaw in nursing care delivery [49]. Essentially, the respectful delivery of nursing care is closely linked to the delivery of patient-centered care [50].

## 5. Research Limitations

It is worth highlighting the conditions resulting from the pandemic caused by SARS-CoV-2, which led to the social isolation of many cancer patients due to their increased vulnerability, making it harder to assess the dimension “Feeling Supported”. Furthermore, we would also like to emphasize the span of the instrument—34 items may be strenuous for some specific populations.

## 6. Conclusions

The study obtained results that indicate that the QONCS_PT scale can be used safely, since it fulfilled all the methodological steps systematically and presents good results regarding its validity and reliability.

We consider that the research conducted is relevant to the fields of oncology, oncology nursing, and nursing management, enabling nurse managers to promote patient satisfaction and thus empowering them. In this sense, our aim is for the QONCS_PT to be widely used in the Portuguese social context, with the ambition of continuously improving the oncology clinical practice of nurses in Portugal. The QONCS_PT has very good psychometric properties, and its use will lead to a continuous improvement in the quality of oncology nursing care and in patient, nurse, and organizational outcomes.

This instrument allows nurses to assess patients’ self-perception of the quality of nursing care provided in an oncological context.

We believe that the instrument will be extremely relevant and useful for nurse managers, since they will be able to obtain a multidimensional perception of the quality of nursing care provided and, inherently, direct interventions that can increase or maintain the quality of care provided, such as training in a professional context, in articulation with intra-hospital resources, or in direct communication with the client’s relatives. These interventions will culminate in greater satisfaction and, intrinsically, an improvement in the quality of care provided in a holistic way, which will certainly contribute to greater cost-effectiveness in the care provided to patients.

## Figures and Tables

**Figure 1 cancers-16-00859-f001:**
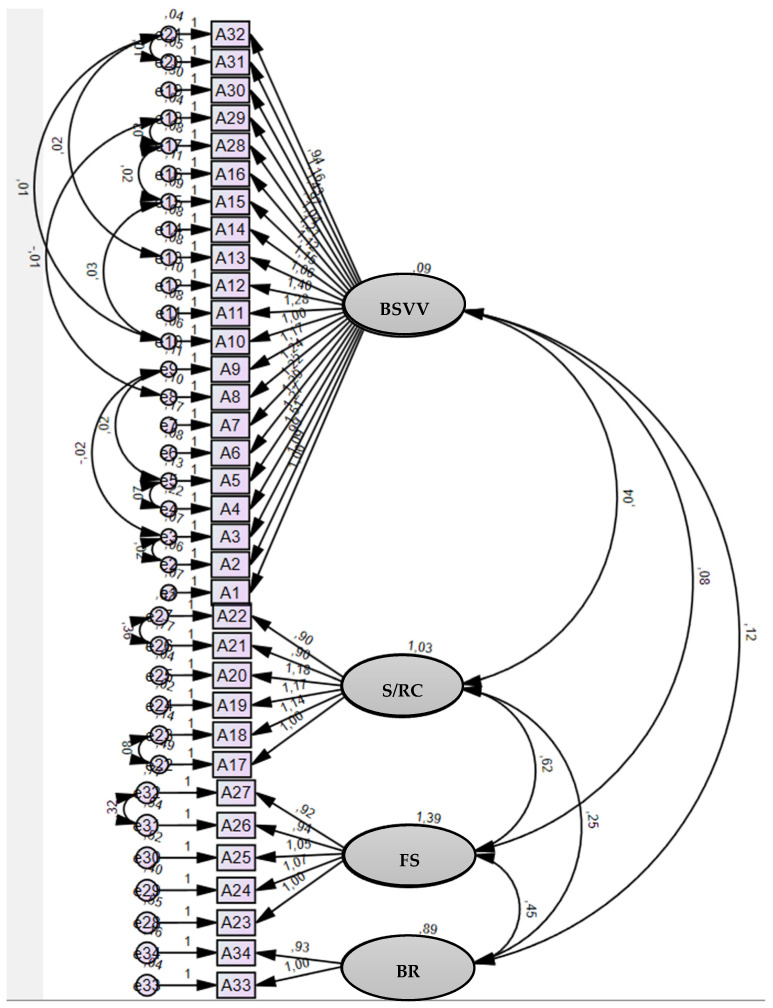
Factorial model of QONCS_PT.

**Table 1 cancers-16-00859-t001:** Sociodemographic characteristics.

Variable	Percentage (%)
Sex	
Male	44.6
Female	55.4
Area of Residence	
Urban	75.7
Rural	24.3
Age	
21 to 28	5.5
29 to 36	5.5
37 to 44	8.4
45 to 52	16.3
53 to 60	21.8
61 to 68	18.3
69 to 76	14.9
77 to 84	6.9
85 to 92	2.5
Diagnosis time (in months)	
<1 month	16.8
1 to 2 months	23.3
>2 months	59.9
Type of cancer	
Breast	22.8
Prostate	1.0
Bladder	1.5
Head and neck	31.7
Lung	4.0
Skin (melanoma)	5.4
Soft tissue	5.0
Other	28.7
Duration of treatment (in days)	
0 to 7	69.8
8 to 14	12.9
15 to 21	9.4
22 to 29	1.5
>29	6.4
Received care for the first time	
Yes	28.7
No	71.3
Received previous care at that hospital	
Yes	64.4
No	35.6
Education level	
Fourth year (primary education level)	20.8
Sixth year (basic education—second cycle level)	9.9
Ninth year (basic education—third cycle level)	15.8
Eleventh year	2.5
Twelfth year (secondary education level)	23.8
Bachelor’s degree	3.0
Licentiate degree	17.3
Other	6.9

**Table 2 cancers-16-00859-t002:** QONCS_PT componentes.

Items	Being Supported, Validated, and Valued	Spiritual/ Religious Caring	Feeling Supported	Being Respected
1	0.763			
2	0.798			
3	0.763			
4	0.695			
5	0.747			
6	0.810			
7	0.678			
8	0.764			
9	0.732			
10	0.796			
11	0.808			
12	0.805			
13	0.774			
14	0.795			
15	0.789			
16	0.748			
28	0.796			
29	0.836			
30	0.604			
31	0.853			
32	0.844			
17		0.857		
18		0.934		
19		0.942		
20		0.936		
21		0.771		
22		0.785		
23			0.780	
24			0.846	
25			0.861	
26			0.824	
27			0.805	
33				0.866
34				0.882
Cronbach’s alpha	0.96	0.96	0.93	0.94

## Data Availability

All the collected forms, as well as databases, were destroyed following the completion of the study and article publication, as agreed in the written informed consent.

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
