# Peer review of "Cross-Cultural Validation of the Portuguese Version of the Quality of Oncology Nursing Care Scale"

_cancers, 2024, doi:10.3390/cancers16050859_

Round 1
Reviewer 1 Report
Comments and Suggestions for Authors
General comments:
1. The paper presents a methodological study on the cross cultural validation of the Portuguese version of the Quality of Oncology Nursing Care Scale.
2. The research topic is relevant for research and practice.
3. The manuscript is well written and understandable.
Specific comments:
4. The tetra-factorial model explained a total variance of the instrument of only 69.8%. This is quite low. Have solutions with more factors a better explanation of the data?
5. A validation of the factor structure identified could be evaluated with another validation sample to scrutinize the observations and conclusions made.
6. Reliability over time needs also be tested.
7. It needs a stronger focus on external validations (e.g., correlations with similar constructs).
8. The cost-effectiveness ratio could be discussed.
9. Potential biases and understanding problems could be discussed further.
10. Is the instrument rather a tool to assess therapy quality or merely satisfaction? Both aspects are different constructs.
Reviewer 2 Report
Comments and Suggestions for Authors
Dear authors,
Many thanks for submitting your work to the journal. I read a well-written and organized manuscript. However, I have made some comments that should be addressed.
Best regards
